# Balancing Memorization and Generalization in RNNs for High Performance Brain-Machine Interfaces

**Joseph T. Costello, Hisham Temmar, Luis H. Cubillos, Matthew J. Mender,**
**Dylan M. Wallace, Matthew S. Willsey, Parag G. Patil, Cynthia A. Chestek**

*Departments of Electrical and Computer Engineering, Biomedical Engineering,*
*Robotics, and Neurosurgery, University of Michigan, Ann Arbor, MI, USA*

## Abstract

Brain-machine interfaces (BMIs) can restore motor function to people with paralysis but are currently limited by the accuracy of real-time decoding algorithms. Recurrent neural networks (RNNs) using modern training techniques have shown promise in accurately predicting movements from neural signals but have yet to be rigorously evaluated against other decoding algorithms in a closed-loop setting. Here we compared RNNs to other neural network architectures in real-time, continuous decoding of finger movements using intracortical signals from nonhuman primates. Across one and two finger online tasks, LSTMs (a type of RNN) outperformed convolutional and transformer-based neural networks, averaging 18% higher throughput than the convolution network. On simplified tasks with a reduced movement set, RNN decoders were allowed to memorize movement patterns and matched able-bodied control. Performance gradually dropped as the number of distinct movements increased but did not go below fully continuous decoder performance. Finally, in a two-finger task where one degree-of-freedom had poor input signals, we recovered functional control using RNNs trained to act both like a movement classifier and continuous decoder. Our results suggest that RNNs can enable functional real-time BMI control by learning and generating accurate movement patterns.

## 1 Introduction

Brain-machine interfaces (BMIs) have the potential to restore motor function to people with paralysis. While intracortical motor BMIs have successfully enabled paralyzed human patients to control computer cursors [1], robotic arms [2], or even move their own muscles through functional electrical stimulation [3], the decoding algorithm that predicts intended movement from neural signals still lacks accurate and naturalistic movement outputs. Recently, non-linear decoding algorithms using artificial neural networks have demonstrated higher offline kinematic prediction accuracy compared to linear approaches, which could enable more functional BMIs. Glaser et al. 2020 found that recurrent neural networks (RNNs), specifically the long short-term memory (LSTM, [4]) and gated-recurrent unit (GRU, [5]), outperformed other decoding architectures for predicting movement in several brain regions [6]. Other neural networks have improved decoding accuracy by de-noising neural signals using RNNs (LFADS, [7]) or transformers [8]. However, offline accuracy does not necessarily predict real-time closed-loop (online) performance [9, 10], where errors are cumulative and the user can adjust neural activity based on feedback. With goals of improving function of real-time BMIs, it is unclear if the high offline accuracy of RNNs translates to the online setting.

In online prediction settings, BMIs using RNNs for continuous movement prediction have demonstrated limited success but have not been evaluated against other neural network architectures. Sussillo et al. 2016 showed that an online RNN was more robust to electrode perturbations and slightly outperformed a linear Kalman filter in a reaching task [11]. In a closed-loop simulation, Hosman et

al. 2019 found that an LSTM decoder could outperform a Kalman filter in a cursor-control task [12]. Recently, Deo et al. 2023 found that RNNs tested with a human subject tended to significantly overfit to offline data and required novel dataset regularization for functional online use [10]. Alternatively, our group found that an ReFIT convolutional feedforward network outperformed a ReFIT Kalman filter in an online two finger group task by accurately predicting a large dynamic range of velocities [13]. Thus, there is a need to evaluate the online performance of RNNs against other high performing neural networks where all decoders have the possibility of overfitting to training data.

In addition to high prediction accuracy, RNNs have the capacity to memorize and generate realistic sequences with minimal input [14, 15, 16]. In the BMI context, an RNN decoder could memorize discrete postures and stereotyped movements between postures, and could help generate movement for degrees-of-freedom (DoFs) where little information is contained in the input (for example, if few cortical channels are tuned to one movement direction). RNNs have previously been used to generate smooth, multi-DoF limb movements [17, 14, 18], and other work has shown that RNNs can learn hundreds of separate simple tasks [19, 16]. However, it is unclear if a memorized RNN is controllable in a closed-loop setting where overfit decoders may easily fail [10].

Here, we evaluate the online closed-loop performance of RNNs against other neural network architectures in non-human primates performing a multi-DoF finger task. We find that offline accuracy generally predicts the relative ordering of online decoder performance, with LSTMs outperforming other architectures. Then, we show how reducing task complexity and allowing memorization of movements can enable the BMI to reach able-bodied performance, with a gradual transition between discrete and fully continuous movement outputs. We provide an example of using selective movement memorization to recover performance despite having poor neural inputs for a particular DoF. Finally, we find that RNNs trained to memorize a small set of movements still rely on user inputs and have some limited capacity to generalize when controlling velocity rather than position.

## 2 Methods

### 2.1 Microelectrode Array Implants

We implanted two male rhesus macaques (Monkeys N and W) with Utah microelectrode arrays (Blackrock Neurotech) in the motor cortex using the arcuate sulcus as an anatomic landmark for hand area, as described previously [20, 21]. In each animal, a subset of the 96 channels in M1, with threshold crossings morphologically consistent with action potentials, were used for offline recordings and closed-loop BMI control. Surgical procedures were performed in compliance with NIH guidelines as well as our institution's Institutional Animal Care & Use Committee and Unit for Laboratory Animal Medicine.

### 2.2 Signal Processing and Feature Extraction

We recorded 96-channel Utah array data using a Cerebus neural signal processor (Blackrock Neurotech). The Cerebus sampled data at 30 kHz, applied a 300-1000 Hz bandpass filter and downsampled to 2 kHz before streaming the data to a computer running xPC Target version 2012b (Mathworks), which calculated spiking band power (SBP) by taking the magnitude of the signal and summing into 32 ms bins. We used SBP since it is highly correlated with neural firing rate and a high-performance BMI input feature [22]. During online experiments with neural network decoders, the binned SBP of each channel was normalized to have zero mean and unit variance, based on the mean/variance of initial training trials.

### 2.3 Behavioral Task for Finger Movement Decoding

We trained monkeys N and W to acquire virtual targets with virtual fingers shown on a computer screen in front of the animal (Figure 1a). In hand control ("offline" trials), monkeys moved their fingers within a manipulandum that measured the angles of the index and middle-ring-small (MRS) finger groups using bend sensors, and controlled the virtual fingers (Figure 1b). During brain control (also known as closed-loop, "online" trials), monkeys controlled the virtual fingers using neural signals and a decoder. The finger task required placing the virtual index finger and MRS fingers on the respective targets and holding for 750 ms during offline trials or 500 ms during online testing. The target size was 15% of the active range of motion. The total number of possible target locations

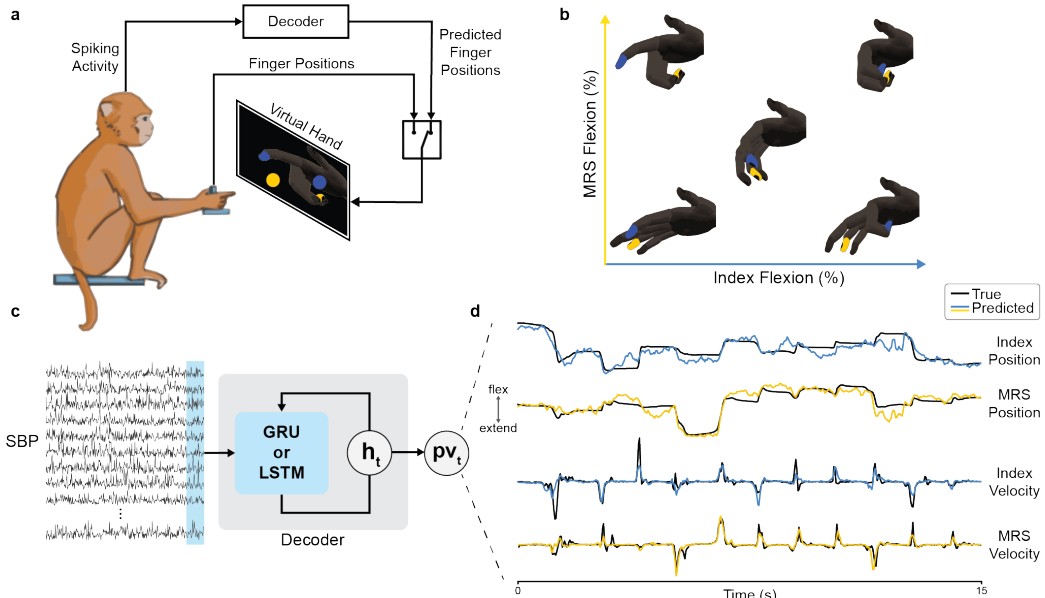

**Figure 1: BMI for decoding finger movement.** (a) Experimental setup. During hand-control trials the monkey used a finger manipulandum to control a virtual hand and acquire targets, with visual feedback from the screen. Spiking-band power (SBP) was calculated in real-time from 96-channel Utah microelectrode arrays. During brain control, the decoded finger positions controlled the virtual hand. (b) Example movements for the 2-DoF task involving flexion and extension of the index finger and MRS fingers. In a random target task, the target is randomly chosen between extension and flexion for each DoF. (c) SBP was averaged into 32 ms bins and fed into an RNN (GRU or LSTM) decoder which predicted position and velocity for one or two finger groups. Other decoders not depicted here were also tested. (d) Example true and predicted positions and velocities from an LSTM decoder for Monkey N performing a 2-DoF random target task. The LSTM is able to predict both slow and fast velocities.

varied based on the task, ranging from 2 targets up to fully random (any position could be chosen), with 1-DoF (just index finger) or 2-DoF (index and MRS fingers). Target locations were randomly chosen at the start of each trial. We collected 500 offline trials for decoder training (although only 300 were collected for offline memorization, Figure 3a). We note that Monkey N typically continued to move his hand even during brain control mode. In this work we only used offline historical data from Monkey W who did not perform online control.

## 2.4   Simulated Datasets

For some analyses we created simulated offline datasets of a virtual user performing the same target acquisition task. The goal of these simulations was to test the relative impact of amount of training data, number of DoFs, and number of inputs on decoder performance, rather than measuring absolute performance. The simulated user moved with a velocity proportional to the distance to the target along each DoF. Artificial neural channels were generated such that each channel had a random relationship with the position, velocity, and acceleration at each timestep, as suggested in [23]. Further details and specifics can be found in Supplementary Methods.

## 2.5   Performance Metrics

Offline performance was measured by the Pearson correlation and mean-squared error (MSE) between predicted and true finger position and velocity. Offline velocities and positions were normalized to have unit standard deviation and zero mean (normalized using training data). We also calculated the mean-absolute error (MAE) between velocity predictions and the true velocity. To do so, true velocities were binned into 21 bins between -3 to 3 standard deviations, and the mean-absolute error was calculated for the predictions of each bin, where data across days were grouped together.

Online performance was measured using two sets of metrics: time and information throughput. Time metrics included the trial-time (the time for a trial with hold time subtracted), move-time (the time until all fingers touch the target for the first time), and orbiting time (trial-time minus move-time, representing extraneous movements where fingers left the target). Information throughput, measured in bits per second (bitrate), has previously been used to measure finger task performance [13] and was calculated using Fitt's law which accounts for both task difficulty and completion time:

$$throughput = \frac{\sum_k \log_2 \left(1 + \frac{D_k - \frac{R}{2}}{R}\right)}{t_{acq}} \tag{1}$$

where $D_k$ is the distance of the $k$th virtual finger to the center of the $k$th target, $R$ is the target radius, and $t_{acq}$ is time to reach the target. Better decoders (fast and precise) have a higher throughput. Before calculating throughput, we removed initial trials from the performance calculation to account for the decoder learning period before steady state performance, which took approximately 1-2 minutes. When more than 100 trials were performed we removed the first 50 trials, and when fewer than 100 trials were performed (typically when the performance was poor and the monkey was unable to complete many trials) we removed the first 20 trials.

## 2.6 Decoders

We tested five decoder architectures, chosen for their high offline or online decoding performance in other studies. Decoders took in binned SBP features (neural features) from up to 96 channels with 1-5 bins of time history for each channel and were trained to predict the position and velocity of 1 or 2 finger groups (Figure 1c). The neural network decoders were implemented and trained using PyTorch 1.12.1. Each decoder is detailed below; see Supplemental Figure 1 for more details on each network architecture. The total number of neural network decoder parameters varied from 479k for the LSTM to 923k for the TFM (see supplemental results).

**Recurrent Neural Network (RNN)**    We tested two variants of RNNs: the LSTM [4] and GRU [5], which have previously shown high performance in offline neural decoding [6]. Decoders were implemented using the torch.nn.LSTM and torch.nn.GRU classes. As shown in Figure 1c, the RNN decoder takes in the current time bin of SBP features and the previous hidden state, updates the hidden state, and predicts position and velocities as a linear function of the hidden state. During training, a sequence length of 20 and zero-initialization for the hidden state were used for each sample. The final output of each sample sequence was used to compute the loss. During online decoding, the hidden state was stored in memory, such that the inputs at each timestep were the current neural bin and the previous hidden state.

**Convolutional Feedforward Neural Network (FNN)**    We tested the FNN described in Willsey et al. 2022 [13], which previously outperformed a ReFIT Kalman Filter [24, 21] in online testing. The network uses an initial convolutional layer over time (for each channel) followed by several feedforward layers that use batch normalization, dropout, and ReLU. The network used bins of time history per channel (160 ms). Implementation details are described in [13] and Supplemental Figure 1, and hyperparameters were optimized as described below. Notably, we did not perform ReFIT recalibration as was done in [13].

**Transformer (TFM)**    Transformers [25] have recently become popular for sequence processing tasks and have performed well in offline neural decoding (Ye 2021). The transformer tested here used a cosine positional encoding layer [25] followed by multi-head attention and feedforward layers, and a final linear layer to output kinematic predictions (see Supplemental Figure 1). The transformer layers were implemented using the torch.nn.TransformerEncoderLayer. The transformer used five bins of time history per channel (160 ms). We note that unlike the transformer in Ye et al. 2021 [8] which was trained to predict smoothed firing rates, the transformer here was directly trained to predict kinematics.

**Kalman Filter (KF)**    The kinematic Kalman Filter (KF) is a linear decoder widely used in BMI that optimally combines a state transition model (using the current position and velocity as a state) with an observation model (that relates the kinematics to neural activity) to estimate the current position and velocity. Like the other decoders, the KF takes in the current bin of neural features and the previous

state, and then outputs predicted finger velocities. The KF was implemented and trained as described in [20, 26] using the NumPy library, but here we did not use ReFIT recalibration.

## 2.7 Neural Network Decoder Training

For offline decoders, datasets were split into 70% for training, 10% validation, and 20% testing. For online decoders, datasets were split into 80% training (400 trials) and 20% validation (100 trials) since a test set was unnecessary. Neural features and finger positions/velocities were normalized based on the training set to have zero mean and unit variance. We first performed hyperparameter optimization for each decoder using two offline datasets for each monkey individually. Bayesian optimization was performed using the Optuna library, and decoder performance was generally robust to the specific choice of hyperparameters. See Supplemental Methods for details and Supplemental Table 1 for parameter values used. Decoders were trained using minibatch gradient descent with the PyTorch Adam optimizer. The loss function was the mean-squared error (MSE) loss with an additional term to penalize finger co-dependence (see Supplemental Methods; the additional term was only used for online decoders). The learning rate was held constant until validation loss plateaued, then halved and held constant until validation loss plateaued a second time (using the ReduceLROnPlateau scheduler). After training was complete, a linear regression was calculated to transform the normalized outputs back to the initial position and velocity scale. Training took less than 5 minutes for each network.

## 2.8 Online Decoder Testing

During online experiments, decoders were run in Python 3.7 using a dedicated linux computer with an RTX 2070 Super GPU (NVIDIA). After receiving binned neural features from the xPC, the decoder calculated position and velocity predictions, and a final position for each finger was sent back to the xPC to update the virtual hand (see [13] for more details). For the RNN decoders, we found that combining velocity and position predictions to generate the final displayed positions (98% integrated velocity, 2% position) was necessary to prevent the virtual fingers from becoming biased (similar to that of [27, 11]), and we used this for all neural network decoders. For tasks with less than 9 targets we increased the positional contribution up to 50%. During online comparisons, decoders were alternated in an A-B-A or A-B-A-B format, with performance averaged across sets. For the 2D decoder comparison with all 5 decoders, each decoder was only tested once (A-B) due to time. An online decoder test was stopped if the monkey was unable to acquire targets for more than 30 seconds or the monkey stopped attempting to acquire targets.

## 2.9 Training Optimizations

To optimize online performance, in early exploration we tested several machine learning techniques for improving accuracy and generalization (detailed in Supplemental Methods). When the initial online trial success rate was 90% or less, we found the following techniques improved individuated finger control and success rate (SR): modifying the loss function to encourage finger independence (11% increase in SR, 12% improvement in trial time), training on more single-finger movements (38% increase in SR, 27% improvement in trial time), and training on trials with added positional perturbations (11% increase in SR, 12% improvement in trial time). Adding a small amount of noise to the neural data during training (as suggested in [28]) improved average trial times by 6%. Finally, we found that training on more data improved offline accuracy (up through 2000 trials), but with only a small improvement after 500 trials. For online results detailed below, decoders used the modified loss function, additional single-finger movements, and added noise.

# 3  Results

## 3.1  High Performance finger BMI with RNNs

**Offline Decoder Comparison**    Previous work has shown that RNN decoders, specifically LSTMs and GRUs, can outperform other feedforward neural networks for predicting offline reaching movements from intracortical signals [6]. Here, we began by establishing a baseline for how well RNN decoders can decode dexterous finger movements, compared to other neural network architectures. One adult male rhesus macaque, Monkey N, was implanted with Utah arrays in the hand area of the

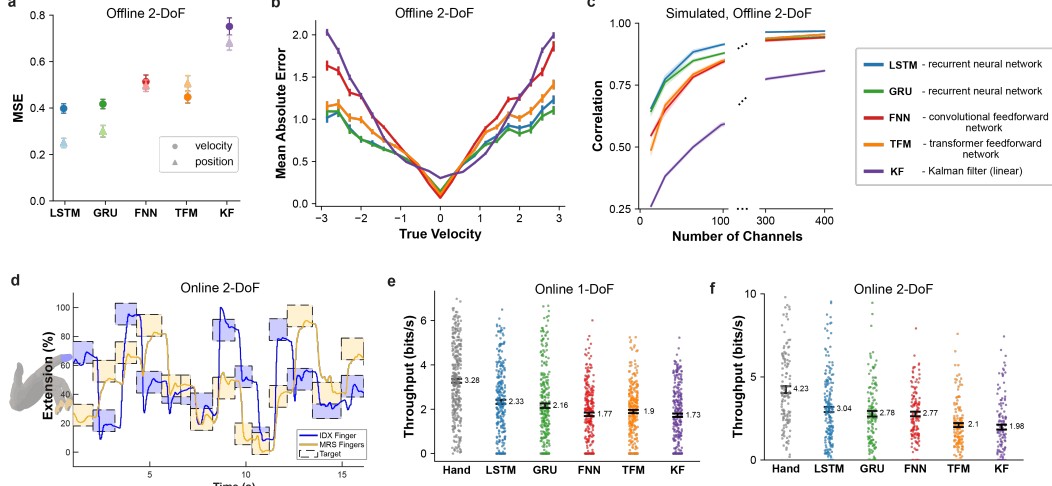

**Figure 2: Offline and online decoder performance comparison.** (a) Offline model performance measured by MSE (a.u) between predicted and actual movements. Values represent the average of ten models trained on ten separate days of the 2-DoF random task. (b) Mean absolute error of velocity predictions where the x-axis is the true velocity and the y-axis is the error in predicted velocity (lower is better). Units are normalized velocity and data is averaged across ten days. (c) Offline correlation between true and predicted movements for simulated datasets with varied channel count. As channel count increases, the difference in accuracy across decoders becomes smaller. (d) A representative online decode trace of Monkey N using an LSTM for a 2-DoF random target task. The fingers quickly move to the target and can hold with minimal overshoot. Boxes represent the target position for each finger and lines represent the finger positions. (e, f) Online model performance on exemplar days where all decoders were tested in the same day for 1-DoF (e) and 2-Dof (f) random tasks. Performance is measured by bitrate (higher is better). "Hand" is the performance during physical hand control (able-bodied performance). Across all figures, error bars denote one standard error of the mean (SEM), and data for a,b,d-f come from Monkey N.

primary motor cortex, and simultaneous neural activity and finger movements were recorded. For a continuous, random target task we compared the following decoders across ten days: LSTM, GRU, FNN, TFM, and KF. An example offline prediction on a representative day using an LSTM is shown in Figure 1d, where the predicted positions and velocities closely match the true finger movements, notably reaching both fast and slow velocities. To evaluate decoder accuracy, we calculated the MSE and correlation between predicted and true finger velocities and positions. As in Figure 2a, for Monkey N, LSTM decoders had the lowest positional and velocity error across all decoders and also the highest correlations of 0.86 and 0.77 for position and velocity respectively. The LSTM had a significant improvement in position and velocity MSE over all decoders (p<0.001 for all comparisons, one-sided paired t-test). All neural network decoders had significantly lower position and velocity MSE than the KF (p<1e-5 for all comparisons, one-sided paired t-test). The same relative ordering of decoder was found when the analysis was performed on historical data from a second monkey (Supplemental Figure 2), albeit with slightly higher errors overall.

To further reveal the difference in decoder accuracy with Monkey N, we calculated the mean-absolute error (MAE) for each ground-truth velocity (Figure 2b). All neural network decoders had similar low error at slow velocities; however, RNNs had significantly lower error for the fastest 10% of velocities than other neural networks (p<1e-5, one-sided t-test), suggesting they may enable faster real-time movements while maintaining slow-movement accuracy. Thus, RNNs more accurately predict offline finger movements than other architectures, agreeing with previous literature [6].

**Online Decoder Comparison**   Since real-time, closed-loop control requires the user to actively react to and correct errors, the online and offline distributions of neural activity and movements may differ, preventing a high performing offline decoder from functioning well online. Therefore, we next compared decoders in a closed-loop setting, where only the monkey's neural activity (rather than his hand) controlled the virtual hand movement. To evaluate online performance, we compared

decoders within multiple sessions with Monkey N, and measured performance using information throughput (bitrate). Decoders were trained using only same-day data. Figure 2d shows example online movements using a 2-DoF LSTM, where the monkey quickly moved each finger group and then held at the target position, for a median time of 1.3 sec per trial. The LSTM tended to overshoot index-flexion targets, which Monkey N also tended to do during hand-control. On all tests, the LSTM decoder matched or had the highest information throughput, with a median bitrate 15% higher than the GRU across 2 comparison days, and 18% higher than the FNN across 3 days. Supplemental Video 1 shows example usage of each decoder.

We next tested if online decoder performance follows the same ordering as offline accuracy. Figure 2e/f shows online performance on two exemplar days where all five decoders were tested in a 1-DoF and 2-DoF task with Monkey N, where performance followed the same ordering as offline accuracy for both comparisons (except for the TFM outperforming the FNN on the 1-DoF task). Interestingly, the FNN had only slightly lower online 2-DoF bitrate compared to the RNN decoders despite significantly lower offline position and velocity correlations. This suggests that offline performance on the same dataset may give a general idea about online performance ordering but does not indicate the specific performance differences during closed-loop control.

**High Channel-Count, Simulated Decoder Comparison**   With the recent developments of higher channel count recording systems [29, 30], BMIs may soon have access to more input channels. To determine if recurrent architectures may be more accurate than feedforward architectures at higher channel counts, we trained decoders on simulated datasets with a varied number of artificial neural channels. As seen in Figure 2c, for 10-100 channels, the LSTM and GRU decoders have significantly higher offline correlation compared to FNN, TFM, and KF decoders (average LSTM correlation 0.1 higher than TFM). However, at higher channel counts the advantage of the LSTM and GRU shrinks, such that the difference in correlation at 400 channels is only 0.01 between LSTM and TFM. While actual neural data may differ from the simulated datasets used here, these results suggest that the choice of a specific nonlinear decoder may be less important at high channel counts.

## 3.2   Able-bodied performance by memorizing movements

As opposed to the fully randomized target postures in the previous section, we next investigated RNN decoder performance on more simple, stereotyped tasks with a limited set of targets. RNNs have the ability to memorize and reproduce learned patterns [14]; here, we aimed to let the RNN memorize movement patterns for a discrete set of targets, potentially allowing for more accurate decoding despite the potential for reduced generalization on a wider range of tasks. To test the capacity for RNNs to memorize targets, we trained and tested LSTM decoders on discrete target datasets, ranging from 2 targets (1D) up to 23 targets (2D). We note that the target order was randomized, which required the decoder to rely on neural input. Separate decoders were trained for each number of targets. We found that LSTM decoders could memorize the specific positions and movements between targets and that offline error increased as the number of targets increased (Figure 3a). Intuitively, this makes sense because as more targets are added, there are fewer example movements per target when keeping the number of trials constant. In additional simulations, we found that as more targets are added, more inputs or training data are needed to maintain accuracy (Supplemental Figure 3). While offline error was highest for random targets, this maximum value corresponds to relatively accurate predictions (as shown in the previous section).

We found a similar trend when testing decoders online with simple tasks (Figure 3b). For tasks with 2 to 4 targets, monkey N using an LSTM decoder achieved 2% higher bitrates than able-bodied manipulandum control on average (see Supplemental Video 2), where the performance improvement was due to a reduced orbiting time (in the 2-target case on average, brain control had a 20 ms slower move time but a 96 ms faster orbiting time). Move time is the time to reach the target and orbit time is time for extraneous movements after reaching the target (see Methods). As the number of targets increased, brain-control move times stayed approximately the same while orbiting time increased (Figure 3c). This suggests that with fewer targets, the RNN learns stronger internal dynamics to minimize extraneous movement. Overall, as more targets are added, thus decreasing offline performance, online performance drops down to approximately random-target performance. Therefore, by changing the training dataset the RNN decoder can smoothly transition between memorization with strong dynamics (similar to a classifier) and generalization with weaker dynamics.

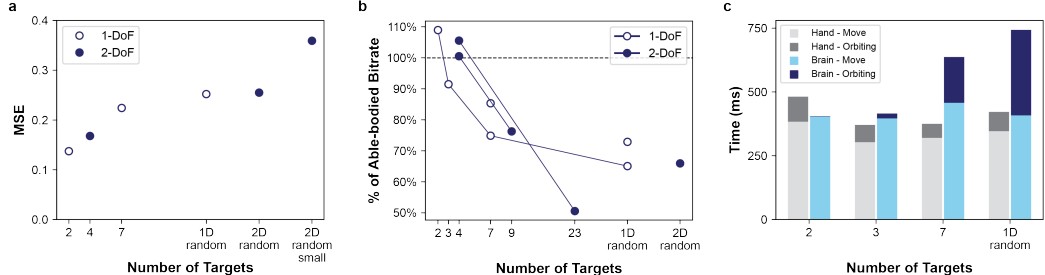

**Figure 3: LSTM decoders have increased performance on simple tasks.** (a) Offline MSE (a.u.) for LSTMs trained on tasks with varied numbers of targets and 1-2 DoFs, where all tasks were performed on the same day. "2D random small" refers to random targets with 25% smaller size, requiring finer control. (b) Online performance for varied numbers of targets and DoFs across multiple sessions, with performance measured relative to able-bodied control (calculated as online-bitrate / hand-bitrate). Lines indicate tests performed within the same day. (c) Average move and orbiting times for hand control (gray bars) and brain control (blue bars) for 1-DoF tasks on an exemplar day. Orbiting times tend to increase with task complexity. Note that the 500 ms hold time is not depicted.

## 3.3 Recovering performance through memorization

In some cases, information on a particular DoF is limited (for example if a BMI has too few channels tuned to that DoF or movement) making the overall BMI unusable. To recover functional use, we hypothesized that the poor DoF could rely on memorization of a discrete movement set (for greater accuracy) while the other good DoF(s) maintain fully continuous movements. For example, Figure 4 shows the first three principal components of the hidden state of a 2-DoF GRU trained on two targets for index finger and random targets for MRS fingers (simulated data, see Supplemental Methods). In this setting, one DoF acts like a movement classifier jumping between line attractors, while the other DoF is continuous with positions continuously represented along each line attractor.

We tested this strategy with Monkey N, who, over a period of 6 months, had a reduction in control of the index finger (likely due to a reduction of neural channels tuned to index finger). When using an LSTM trained on a 2 finger group (2-DoF) random task, index finger had poor online control, resulting in a success rate of 66% (average bitrate 1.14, 112 trials). To recover performance, on the same day, we trained an LSTM on a modified task with only three targets for index finger (extend, center, flex) while maintaining random targets for MRS fingers. By reducing the number of distinct index finger movements, the number of training examples per movement is increased, allowing the RNN to learn more accurate movement dynamics. The LSTM trained on the modified task had substantially higher online performance (success rate of 98%, average bitrate of 4.18, 262 trials) and the finger groups were visibly more independent (see Supplemental Video 3). Thus, for BMI or prosthetics users with poor input quality or limited data, functional online performance may be recovered by modifying the decoder training dataset without necessarily tuning hyperparameters or parameters of the decoder itself.

## 3.4 Memorized RNNs partially generalize to continuous tasks

For RNNs trained to memorize simple postures, it is unclear how well these decoders can generalize to other tasks. One might expect that, when trained to output a discrete set of positions, the RNN may learn strong fixed points (or other dynamics) that enable the RNN to accurately output discrete positions, but may perform poorly when tasked with holding intermediate postures (random targets). To test this assumption and the ability of memorized RNNs to generalize, we trained LSTMs on tasks with a limited number of targets and tested the decoders on online random target tasks. We tested the generalization ability of positional (50% position, 50% integrated velocity) and velocity (1% position, 99% integrated velocity) variants of the same decoder (Supplemental Figure 4).

LSTM decoders trained on 1-DoF 2- and 3-target tasks had 100% and 98% online success rate (SR) respectively when tested on the same simple tasks (positional variant). When tested on the random target task, as expected, the positional decoders had difficulty stopping on targets that did not overlap with the learned targets, and failed to move outside of the learned flexion-extension range. The

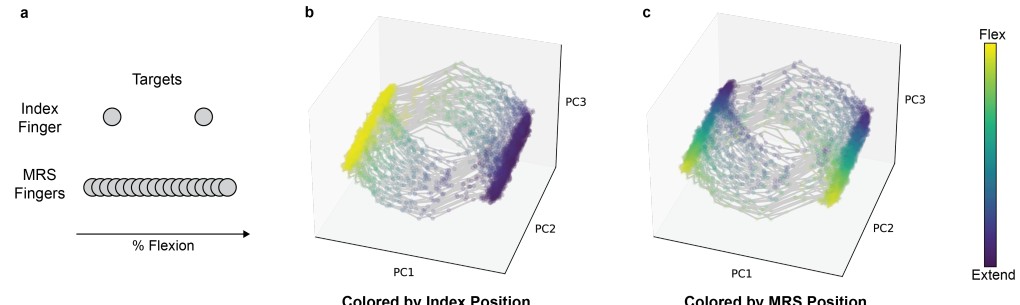

**Figure 4: Hidden states of a memorized RNN.** (a) A GRU RNN was trained on a modified task with random targets for MRS fingers but only 2 targets for index finger (simulated data). (b, c) Plots depict the first three principal components of the RNN hidden state, with traces indicating the path over time. (b) and (c) plot the same data, just with different coloring. When the path is colored by the predicted index position (b), it is clear each line attractor corresponds to an index target. As in (c), the position along each line attractor corresponds to the predicted MRS position. Thus, the RNN can learn different dynamical structure for each DoF to act more like a classifier when beneficial.

2-target positional decoder was non-usable (SR < 50%), while the 3-target positional decoder was more functional (SR 87%, bitrate 1.05). Interestingly, the velocity decoder variants were better able to generalize, where the 2-target velocity decoder had a 98% SR (bitrate 1.33) and the 3-target velocity decoder had an 80% SR (bitrate 0.93, lower performance likely due to low monkey motivation). While functional, the velocity variants had more difficulty making small corrective movements and had lower bitrates compared to random-trained decoders (bitrate 2.19). These results suggest that memorized RNNs can generalize to some degree, where training on more targets leads to better positional generalization.

## 4   Discussion

A key component of BMIs that restore motor function is an accurate decoding algorithm that performs well in online settings. Unfortunately, for continuous movement decoding, offline accuracy does not incorporate closed-loop dynamics, meaning offline metrics may fail to predict online performance [9] and offline cross validation does not fully reveal decoder overfitting [10]. However, here, using a multi-DoF finger BMI, we found that offline accuracy generally predicted the relative online decoder performance ordering. We found that recurrent neural networks, namely LSTMs, outperform other neural network architectures in both offline and online settings. This high performance of RNNs follows previous work showing that decoders that learn neural or kinematic state dynamics (including RNNs) can more accurately predict kinematics from noisy neural data [31, 27, 6]. All the networks tested here were relatively small (5 layers or fewer) and could be efficiently run in real time (typically <2 ms per prediction), suggesting that very deep or complex architectures may not be necessary for high performance. Additionally, we did not perform intention-based recalibration (ReFIT; as described in [24, 13]) due to experimental trial count constraints. This could potentially benefit some decoders more than others. Further work may also investigate the performance trade-offs of decoders trained to de-noise neural data [7, 8] versus decoders directly trained to predict kinematics.

While the neural network decoders tested here were all highly usable with minimal apparent overfitting, Deo et al. [10] found that RNNs tended to significantly overfit to offline data when tested in humans. In humans with paralysis, decoding algorithms are trained from offline data by instructing participants to attempt movements in sync with an artificially moving effector. With able-bodied users (like the non-human primates used here), however, the direct relationship between neural activity and true movement can be used for training. These able-bodied movements more broadly sample the range of positions/velocities and neural activity needed to perform a task and often have errors like overshooting. When trained on only two able-bodied targets, enough variation was present to allow for generalization to other movements. These results suggest regularization of neural networks is necessary for high performing BMIs, whether inherent to the training data or by applying artificial data modifications.

In additional testing we found that using more time history in the FNN and TFM decoders (up to 20 bins/640 ms) resulted in poor online control (Supplemental Figure 5). This is likely due to overfitting to training data that lacks many of the corrective movements present in online control, and demonstrates how offline analyses can fail to predict the performance in the closed loop setting. Interestingly, the RNNs were trained with longer history but did not display this overfitting. Future work may investigate methods of using longer training sequences while limiting overfitting and lag, for example by discouraging attention to early bins.

Decoder memorization of training data is commonly associated with overfitting, suggesting poor closed-loop control. Here, we found that RNN decoders allowed to memorize a few movement patterns were highly controllable online and matched able-bodied performance for tasks with 2-4 target postures. With a reduced target set the RNN can learn stronger recurrent dynamics for higher accuracy, acting somewhat like a pseudo-classifier between movements. As task complexity (the number of targets and distinct movements) increases, offline and online decoder performance degrades gracefully down to fully continuous performance, partly due to the reduced number of training examples per movement. This is similar to neural word decoding, where word error rates increase with larger vocabularies [32]. Unlike algorithms that manually combine a classifier and continuous decoder [33, 34], RNNs are fully continuous yet can internally act as a classifier and generator for movements that lack input information, while maintaining high performance continuous output for other movements. As BMIs extend to more DoFs, our results suggest that more data or neural channels will be needed to maintain accuracy, and the degree of memorization can be controlled for each DoF independently. Future work may investigate the relative dependence on inputs vs autonomous dynamics of RNNs, and their effect on online performance.

There are several limitations to the present study of closed-loop BMI performance. First, we did not explore every possible decoder architecture. More optimal architectures likely exist, but here we found several key factors of each class (for example, the need for limited time history of feedforward models, and the ability of RNNs to strongly learn few movements). Future work may explore models that combine recurrence with feedforward attention layers to take advantage of the benefits of each (for example, [35]). Also, only one monkey was used for the online analyses; additional users may have different control strategies or decoder errors.

Future BMI decoders could be designed to be more explicitly tunable to allow the user to adjust the balance between memorization (with more reliance on internal dynamics) and generalization to a wider range of movements. A decoder could act more like a classifier when producing stereotyped movements such as sign language or functional grasps, but act as continuous controller when producing arbitrary movement. Users at the onset of BMI use could use a simple task decoder that works well with minimal training data (reducing cognitive fatigue and maintaining user motivation), then transition to higher complexity as more data is acquired. These strategies could also be employed for myoelectric and other prosthetic controllers which must function using noisy signals with large uncertainty in user intentions. BMIs that can produce accurate kinematic predictions across the range of few neural channels to thousands could speed the clinical translation of BMIs with current and future hardware.

# 5 Acknowledgements

We thank Eric Kennedy for animal and experimental support. We thank the University of Michigan Unit for Laboratory Animal Medicine for expert surgical and veterinary care. JTC was supported by NSF GRFP under grant 1841052. DMW was supported by Dan and Betty Kahn Foundation (Grant 2029755) and University of Michigan Robotics Institute. HT, MJM and PGP were supported by NSF grant 1926576. LHC was supported by Agencia Nacional de Investigación y Desarrollo (ANID) of Chile. MSW was supported by NIH grant T32NS007222. CAC was supported by Dan and Betty Kahn Foundation (Grant 2029755) and NSF grant 1926576.

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
