# OpenReview forum: "Balancing memorization and generalization in RNNs for high performance brain-machine Interfaces"
_NeurIPS.cc/2023/Conference — NeurIPS 2023 spotlight_

### Official Review · Reviewer_ZEFp · 2023-06-11

**Soundness:** 3 good
**Presentation:** 4 excellent
**Contribution:** 4 excellent
**Rating:** 8
**Confidence:** 4

**Summary:**

This paper explores the utility of different classes of neural decoders for online brain machine interfaces. The authors first show that recurrent network decoders outperform transformers, CNNs and Kalman filters in an offline setting. They then validate that the superiority of RNN-based decoders also holds in the online setting. Finally, the authors show that such RNNs can flexibly interpolate between memorizing output trajectories in a few-target setting and integrating instantaneous neural activity in a many/continuous-target setting, and that this can be done independently for different actuators.

**Strengths:**

Investigating the consistency or discrepancy of different neural decoders between offline and online BMI settings is a very important question in modern systems neuroscience. This is especially true since many labs and researchers do not have the capacity for online BMI experiments, and we as a field therefore strongly rely on offline metrics for motivating different neural decoders.

The question of memorization in a few-target setting vs. flexible control in a many-target setting is also interesting, and the analyses are clear and informative.

In general, this is a paper that asks interesting questions, answers them with convincing data and analyses, and is well written and presented, and I very much enjoyed reading it.

**Weaknesses:**

A major difficulty with studies like this is to tease apart the _potential_ utility of different types of decoders/architectures from the _specific implementation choices_ made. Not much work is done to explore the range of speed/accuracy tradeoffs when e.g. varying network or context size (see below), leaving the reader with only a single data point per neural architecture to draw conclusions from.

**Questions:**

I generally liked the paper, but here are a few suggestions/questions that I thought it would be helpful to clarify if the authors have the capacity to do so.

L40-41: The authors seem to assume that the reader knows what a 'ReFIT' network is, but this would be useful to explain in half a sentence.

Eq 1: I'm not sure I fully understand how information throughput is computed, since t_acq is a single number for a given trial, while D_k continuously changes throughout a trial. Which D_k is used? and does the sum only run over fingers constrained by the task (equivalent to setting R=infinity for non-constrained fingers)?

Section 2.5: Since the RNNs have an adaptive memory and can (in theory) learn to retain information over as long timescales as are necessary for solving the task, it might be interesting to explore different convolutional filter sizes for the CNN and different context windows for the transformer. Is the lower performance of these models because you need more than 160 ms of history to accurately predict movements? or is it something more fundamental? and if you can recover RNN-level performance with more temporal history, what is the resulting cost in computation speed? One could imagine a plot that shows performance against time-per-forward-pass for varying context sizes for both the CNN and transformer, and comparing this to the performance/speed of the LSTM/GRU networks.

Related to the above, it would be good to report the total number of parameters for each network, and potentially the time-per-forward-pass, in the main text, even if the full architecture details can be found in the appendix.

Fig 2: It would be nice to show online performance in terms of trial time in addition to information throughput, since that is a much more intuitive metric.

L235: I found it confusing that this section came after the online decoding section, which is also inconsistent with the layout of Figure 2. I was also wondering whether a similar analysis could be done by subsampling the experimental data instead of using simulated data? On the whole, I found this analysis slightly tangential to the rest of the paper and think it could be moved to the supplement if the authors are low on space.

Figure 3: Is there a way to more directly investigate the dependence of the RNN decoders on inputs vs autonomous dynamics? e.g. removing inputs part way through the trial and investigating the effect on performance for different conditions? Or comparing the relative magnitude of 'recurrent' inputs and 'external' inputs across time and conditions?

Section 3.4: It would be useful to put these results in a figure (and potentially place it in the supplement if there is no room in the main text). For example, it would be informative to have a plot that shows performance as a function of target extension (in the continuous test setting), with the discrete training data points highlighted, to visualize how performance decreases away from the training data in the 2-target and 3-target cases.

Supplementary videos: I found these videos informative but it would be helpful to add a section in the supplementary text/methods that provides a brief legend for each video.

Supplementary Figure 3: the title says "LSTM performance" and the legend says "a GRU decoder was trained".

**Limitations:**

The limitations of the study are not discussed to a great extent. Things that may be interesting to touch on (if there is room) could include:

(i) the impossibility of exploring the full space of network architectures within each class (RNN vs CNN vs transformer) and what that means for the conclusions about which is 'best'.

(ii) thoughts about the generalizability of the results to more complicated tasks, e.g. N-DoF tasks or non-motor tasks. Would the authors expect RNNs to also be superior for BMI in brain regions that rely less on autonomous recurrent dynamics during natural behavior?

---

> ### Author Rebuttal · Authors · 2023-08-09
>
> We thank the reviewer for their questions and helpful suggestions.
>
>
> **“ReFIT” is not explained.**
>
> We propose modifying line 40 to briefly explain ReFIT:
> “Alternatively, [30] found that a convolutional feedforward network using using intention-based recalibration (ReFIT) outperformed a ReFIT Kalman filter in an online two finger group task by accurately predicting a large dynamic range of velocities.”
>
>
>
> **How is information throughput calculated for each trial since D_k continuously changes? (Eq. 1)  How many fingers does the sum include?**
>
> Information throughput was calculated using the distance between finger and target at the start of the trial, so D_k has only one value per trial. We propose modifying line 108 to say “where D_k is the initial distance of the kth virtual finger to the center of the kth target…”. This calculation ignores fingers that are not being actively controlled, for example, in the 2-DoF task only 2 fingers are included in the calculation and all other DoFs are ignored.
>
>
>
> **Is the lower performance of the feedforward models due to the lack of time history?**
>
> We found that adding more time history resulted in better offline prediction accuracy but worse real-time performance. See the global response for detail.
>
>
>
> **It would be good to report the total number of parameters for each network, and potentially the time-per-forward-pass.**
>
> We propose adding the following line to results at line 198, “The total number of neural network decoder parameters varied from 479k for the LSTM to 923k for the TFM (see supplemental results).” And adding a table of parameter counts to the supplement:  “GRU – 639k, LSTM – 479k, FNN – 821k, TFM – 923k, KF – 10k”
>
> All networks performed one decoding step in less than 2ms, such that this processing time is not a limiting factor even if faster screen refresh rates were used (than the 32ms update rate used here).
>
>
>
> **Figure 2: It would be nice to also show trial time in addition to bitrate since it is an intuitive metric.**
>
> While bitrate is not as intuitive as trial time, it takes into account individual trial difficulty as measured by the distance from finger to target at the start of the trial, for a single value that describes performance. However, we propose giving an example of trial time in the results text, line 221, “Figure 2d shows example online movements using a 2-DoF LSTM, where the monkey quickly moved each finger group and then held at the target position, for a median time of 1.3 sec per trial.”
>
>
>
> **Could the comparison of decoders at simulated high channel counts be done by subsampling the experimental data?**
>
> In this analysis we aimed to get an idea of relative decoder performance at higher channel counts. One could imagine taking linear combinations of existing real channels to generate more artificial channels, however, this does not add information. Without introducing more channels with new information (like we can do in simulation), it’s unclear if this technique would be suggestive of decoder performance at high channel counts.
>
> Although the analysis is slightly tangential to the other results, it provides context about how this result may look with future high channel count systems. Here, we found RNNs to outperform other networks, but with future systems this might not be the case.
>
>
>
> **Is there a way to more direct way to investigate the dependence of the RNN decoders on inputs vs autonomous dynamics?**
>
> This is an interesting question that we’ve begun to investigate. As you mention, we can compare the magnitude of the external vs recurrent inputs over time and conditions. We can also create stronger or weaker dependence on dynamics by adding a loss function penalty, and measuring the resulting effect on closed-loop control. Here we focused on functional examples of different decoder architectures, and in future work we plan to explore RNN dynamics more thoroughly.
>
> We propose adding a line to the discussion, line 353, “Future work may investigate the relative dependence on inputs vs autonomous dynamics of RNNs, and their effect on online performance.”
>
>
>
> **Section 3.4: It would be useful to put these results in a figure (and potentially place it in the supplement if there is no room in the main text).**
>
> We propose adding a figure to the supplement with this data, as shown in the attached global response pdf.
>
>
>
> **It would be helpful to add a legend to the supplemental videos.**
>
> We will add a legend in the final version.
>
>
>
> **The impossibility of exploring the full space of network architectures within each class (RNN vs CNN vs transformer) and what that means for the conclusions about which is 'best'.**
>
> We plan to add a discussion on the difficulty of exploring the full parameter space, as noted in the global response.
>
>
>
> **Thoughts about the generalizability of the results to more complicated tasks.**
>
> We address this question in the global response and propose an addition to the Discussion.
>
>
>
> **Would the authors expect RNNs to also be superior for BMI in brain regions that rely less on autonomous recurrent dynamics during natural behavior?**
>
> It is unclear whether feedforward or recurrent models may be superior in brain regions with weaker autonomous dynamics (meaning the region is more feedforward and/or input driven). One might expect feedforward models to predict continuous outputs with similar accuracy to RNNs since there is no benefit to internally modelling the neural dynamics. However, with relatively few or noisy neural channels, there is still the need to infer missing information about the output. In this case, a model (such as an RNN or a KF) that models realistic continuous movements and maintains a state may be more accurately “denoise” the output predictions. For example, a Kalman filter that incorporates a model of kinematics or physics can often outperform a simple linear regression. Further study is needed to investigate this and is out of scope for the current study.

---

> > ### Comment · Reviewer_ZEFp · 2023-08-10
> >
> > I appreciate the thorough responses of the authors to myself and the other reviewers and am glad to see that they have addressed most of the comments highlighted in the reviews.
> >
> > I found it particularly interesting that the transformers/CNNs with longer temporal contexts perform worse in the online setting and am glad that the authors will mention this in the revised manuscript. I would even consider adding a supplementary figure with these results, since a major strength of the paper is the ability to compare online and offline BCI performance and highlight the challenges in translating offline results/methods to the online setting (which is super important since most studies only have the capacity for offline analyses).

---

> > > ### Author Response · Authors · 2023-08-14
> > >
> > > Thanks for the suggestion and noting the strength of the online comparisons. We will add a supplemental results figure, showing a drop in success rate when more than 160ms of history is used for the transformer.

---

### Official Review · Reviewer_cHBb · 2023-06-23

**Soundness:** 3 good
**Presentation:** 4 excellent
**Contribution:** 3 good
**Rating:** 7
**Confidence:** 2

**Summary:**

The work analyzes the performance of different decoders for finger movement prediction from monkeys in different offline and online settings. The authors evaluate recurrent networks, convolutional networks, transformers and Kalman filters. They find recurrent networks to perform best both for online decoding as well as offline decoding. The order of offline performance of the decoders predicts the order of online performance, but the magnitude of differences between decoders varies between offline and online settings. The work also investigates the effects of the degrees of freedom and the number of discrete movement targets (up to an infinite number/continuous movement decoding). Results for recurrent neural networks suggest that varying the number of discrete targets allows to interpolate between the behavior of a classifier and a continuous movement decoder, where for few targets, most of the actual movement to the target is memorized and with more and more targets, more of the movement is decoded from the brain signal.


**Strengths:**

The topic of continuous movement decoders is scientifically interesting and the comparisons  are insightful.
Additional evaluations looking at number of targets etc. are also valuable.
Figures help understand method and results, fairly clean (except a bit blurry Fig 4).

The model types (convolutional, recurrent, transformer, kalman) seem well-chosen to cover typically used decoder types.

There are a lot of subanalyses like higher-channel-count simulated data that help better understand the factors affecting the behavior/performance of the different decoding methods.

The manuscript is well-written and covers many questions and most technical details clearly. Results cover interesting aspects like move and orbiting time.

**Weaknesses:**

Some details could be made more clear, see Questions.

ReFIT is first used without an explanation what it means.

Figure 4 is blurry and hard to read.

A small detail, but making the "Figure X" tests into references, so you can click on them, would be nice.


**Questions:**

During online decoding experiments training part, is the movement 100% controlled from the start by the decoded brain signal? Or is it in any way  slowly transitioning  from  desired movement to decoded movement as some other works do?
What exactly are the simulated data used in 3.1?


"When more than 100 trials were performed, we removed the first 50 trials from the performance calculation to account for a decoder learning period. When fewer than 100 were performed (typically when the performance was poor and the monkey was unable to complete many trials), the first 20 trials were removed from analysis."
Was there some reason to choose exactly 50 and 20?

**Limitations:**

Main results are from a single monkey.

---

> ### Author Rebuttal · Authors · 2023-08-09
>
> We thank the reviewer for their questions and helpful suggestions.
>
>
> **ReFIT is first used without an explanation.**
>
> We propose modifying line 40 to briefly explain ReFIT:
> “Alternatively, [30] found that a convolutional feedforward network using intention-based recalibration (ReFIT) outperformed a ReFIT Kalman filter in an online two finger group task by accurately predicting a large dynamic range of velocities.”
>
>
>
> **Are training movements fully controlled by decoded neural signals?**
>
> In this work, training movements were controlled by the monkey’s physical fingers in a manipulandum, rather than decoded neural signals. Then, after training the decoder, during test trials the movements were 100% controlled by the decoded neural signals.
>
> As you mention, there are methods to gradually transition from idealized movements to full brain control (Velliste et al. 2008), which may be necessary when the user is paralyzed or is amputated. Here, we simply used the able-bodied training movements since the goal was to compare maximal decoder performance rather than training methods. Also, for a fair comparison across decoders, each decoder would need to be trained with the gradual method (taking 5-10 minutes each), which would be infeasible due to the monkey’s limited window of motivation.
>
>
>
> **What is the simulated data in 3.1?**
>
> We appreciate the reviewer noticing this confusing reference.  The simulated data was outlined in the supplementary methods (removed from the main text due to space restrictions), but not referenced in the main text. We propose adding a methods section to describe the simulated data, as explained in the Global Response.
>
>
>
> **What is the reasoning behind the number of initial trials removed from analysis?**
>
> The goal of removing initial trials from analysis was to account for the learning period (when the monkey was getting used to using the decoder) and only measure maximal steady-state performance. The monkey qualitatively took approximately 1-2 minutes to reach this steady-state period, which corresponded to ~50 trials when the decoder functioned well and ~20 trials when the decoder functioned poorly.
>
> We propose the following change to line 110 for clarity:
> “Before calculating average bitrate, we removed initial trials from the performance calculation to account for the decoder learning period before steady state performance, which took approximately 1-2 minutes. When more than 100 trials were performed we removed the first 50 trials, and when fewer than 100 trials were performed (typically when the performance was poor and the monkey was unable to complete many trials) we removed the first 20 trials.”
>
>
>
> **Main results are from a single monkey.**
>
> We agree that only having one monkey is a primary limitation of the study. The specific decoder errors and absolute closed-loop performance would likely vary between monkeys or human users. To bring more confidence to the results, we performed the offline decoder comparison (3.1) using historical data from a second monkey (briefly mentioned, with full results in Supplemental Figure 2) and using simulated artificial neural data (Figure 2c). In all datasets the same relative performance ordering of decoders was found.
>
> For the analyses investigating RNN memorization, the interesting result is that memorized RNNs can be highly functional for closed-loop control despite the possibility to overfit. While one example user (monkey) is enough to show that is high performance is possible, it doesn’t fully explore aspects specific to the user (like absolute model accuracy or performance under different user control strategies).
>
> We propose adding a paragraph to the discussion to mention this limitation, as explained in the Global Response.

---

> > ### Comment · Reviewer_cHBb · 2023-08-11
> >
> > Thanks for answering and clarifying several of these points. It's nice you plan to mention the limitation of the single monkey more strongly, to me still remains the strongest limitation of this interesting and well-written work.
> >
> > An additional point, not necessarily to include in this work but just as general comment about the problems applying convolutional networks for online decoding discussed with other reviewers: Studies that look at the focus of the receptive field of ConvNets may be interesting. For example https://arxiv.org/abs/2010.02178 and similar works. I can imagine here online control  with longer temporal input windows may also be hampered by the fact that ConvNets are most strongly affected by the center of their input.

---

> > > ### Author Response · Authors · 2023-08-14
> > >
> > > Thanks for the helpful reference on convnet receptive fields. We plan to investigate this in future work, especially in the context of closed-loop control where the convolution might be learning an optimal neural lag.

---

### Official Review · Reviewer_WBvg · 2023-07-05

**Soundness:** 2 fair
**Presentation:** 3 good
**Contribution:** 2 fair
**Rating:** 6
**Confidence:** 3

**Summary:**

The authors study the performance of various models (GRUs, LSTMs, TFMs, FNNs, KFs) in reading out neural population activity during one and two finger tasks in non-human primates. Overall, they find that RNNs (GRUs & LSTMs) outperform other models during online tasks and serve as the most promising model for real-time brain machine interfaces (BMI).


**Strengths:**


The paper demonstrates the real-world utility of contemporary machine learning models in BMI. The methodological approach seems sound. Despite the 10-page limitation of the NeurIPS submission, it is clear that a lot of work and effort (both experimentally and data analytically) was put into the making of this study.


**Weaknesses:**

I have one conceptual weakness, and several more methodological weaknesses. The primary weakness is that it’s unclear to what extent these claims will generalize to more interesting BMI tasks – their results already demonstrate the limitations of their experimental designs (i.e., when more movements exist, the capacity of RNNs are effectively reached). It’s also not clear how generalizable the solution of “balancing memory and generalization” is motor BMI, when the ecologically valid degrees of freedom are extremely large.

Methodologically, I am skeptical about how/why transformers perform so poorly. Transformers are powerful precisely because they can learn dependencies in their context window. In the case of a BMI task, this might be temporal neural activity in the preceding time window, with the embedding dimension corresponding to the number of neurons/channels. In this case, why were the transformers limited to only 5 bins of time history per channel (160ms) (line 140-141). This seems extremely limiting, and it is likely that GRUs and LSTMs have a ‘memory’ that is significantly longer than this. A sequence of length 5 does not seem to play to the strengths of a transformer. There are also more recent ‘recurrent’ transformer models that may be relevant to test here.

**Questions:**

What are the practical limitations of extending the claims here to more naturalistic BMIs? Balancing memory and generalization in 1 or 2 DoF tasks is an extremely special case. This should be discussed.

Does the performance of the transformer model improve if a sequence length of greater than 5 (i.e., longer than 160ms) is included in the transformer context window (i.e., the memory of the transformer)?

What are the number of parameters per model? Are they comparable?

Are there corresponding results for Figure 3 for the other models?

I don’t quite grasp the discussion regarding the ability of LSTMs memorizing the positions and movements of targets when number of targets are low. Is this supposed to be a good thing? This seems to me more of a failure/limitation of the model, since it fails to abstract what the neural population is actually doing (controlling finger movements, rather than directing to specific targets).

Line 339: “These results suggest regularization of neural networks is necessary for high performing BMIs”. It wasn’t clear to me which reported result this sentence was referring to.

What are the theoretical reasons as to why RNNs may perform better than other models for continuous BMI?


**Limitations:**

The limitations are mostly in terms of the scalability of a ‘memorization’ strategy to ethological and naturalistic tasks. More generally, it’s not clear how these findings would scale to a more natural task.

---

> ### Author Rebuttal · Authors · 2023-08-09
>
> We thank the reviewer for their questions and feedback.
>
>
> **What are the practical limitations of extending the claims here to more naturalistic BMIs? (More DoFs)**
>
> We address this question in the global response and propose an addition to the discussion. Briefly, more DoFs may require more training data and/or more channels. With more channels (and more information), we might reduce the amount of memorization for more naturalistic control.
>
>
>
> **Does the performance of the transformer model improve if a sequence length of greater than 5 is included in the transformer context window?**
>
> This is a good question since we’re not taking advantage of one of the transformer’s strengths. We found that using longer sequence lengths resulted in better offline accuracy but worse closed-loop control. See the global response for details and a proposed addition to the discussion.
>
>
>
> **What are the number of parameters per model? Are they comparable?**
>
> All the neural network models had the same order magnitude number of parameters, but the LSTM used significantly fewer parameters than the TFM:
> GRU – 639k, LSTM – 479k, FNN – 821k, TFM – 923k, KF – 10k
> Notably, the feedforward models required more parameters than the RNNs. For each model, we chose the layer sizes that maximized offline accuracy, aiming to not overparameterize each model.
>
> We propose adding the following line to results at line 198, “The total number of neural network decoder parameters varied from 479k for the LSTM to 923k for the TFM (see supplemental results).” And adding a table of parameter counts to the supplement: “GRU – 639k, LSTM – 479k, FNN – 821k, TFM – 923k, KF – 10k”
>
>
>
> **Are there corresponding results for Figure 3 for the other models?**
>
> Figure 3, which details the performance on varied numbers of targets, was only performed using LSTMs and not replicated with the other models. While it would be interesting to test other models on the varied tasks, we chose to test the highest performing model (LSTM) due to experimental time and paper length constraints. We hypothesize that similar trends would be found with the other models, albeit with lower performance. Since the feedforward models are limited to only 5 bins of history, one might expect them to have a lesser ability to generate smooth and precise movements compared to the RNNs with a longer time history.
>
>
>
> **Why is memorization of few targets a good thing? This seems like a model failure since the model fails to abstract to what the neural population is actually doing.**
>
> As you mention, “memorization” limits the output space of the model, may prevent generalization to the continuous range of postures, and does not fully abstract the neural dynamics. However, from a BMI controls perspective, we want a controller that can accurately and robustly move between postures to enable functional use, even if this somewhat limits the output capability. For example, someone with perfect control of a prosthetic with a few distinct positions may have a functional advantage over someone with moderate control of a fully continuous prosthetic. When a user has more neural channels with more information, it probably makes sense to use regular continuous control. However, with current implants it is common to have poor signal quality, in which memorization can be useful. Also, it was interesting to find that even when the decoder “memorizes” outputs, it still relies on the user’s inputs vs fully relies on autonomous dynamics (which was initially unclear).
>
>
>
> **Line 339: “These results suggest regularization of neural networks is necessary for high performing BMIs, whether inherent to the training data or by applying artificial data modifications”. It wasn’t clear to me which reported result this sentence was referring to.**
>
> In this discussion paragraph we compare a recent study that found when training on artificial movements in humans, significant data augmentation was required to prevent overfitting. Here, we trained on real movements that incorporate imperfections and we did not have a problem with overfitting. The “regularization” we refer to in line 339 is the natural variation in our training data that may help to regularize the network, resulting in the generally high performance overall (for example, matching able-bodied performance in section 3.2.
>
>
>
> **What are the theoretical reasons as to why RNNs may perform better than other models for continuous BMI?**
>
> Models that maintain an internal state may better “denoise” neural inputs. For example, a Kalman filter that incorporates a model of kinematics or physics can often outperform a simple linear regression. Similarly, RNNs can learn an internal state that helps it understand the neural dynamics and to smoothly generate continuous outputs, despite noisy input. Feedforward models without an internal state may be able to accurately predict continuous outputs when given enough time history. However, as we found in real-time decoding, using long time history results in overfitting and poor performance.

---

> > ### Comment · Reviewer_WBvg · 2023-08-14
> >
> > Thanks to the authors for their clear rebuttal. A few follow-up questions.
> >
> > 1. Why does the GRU have so many more parameters than the LSTM? Does it a have fewer units? Typically GRUs should have fewer parameters than an LSTM of the same size.
> > 2. Isn't it possible that the number of parameters in the feedforward class of models makes it difficult for it to train in the closed-loop setting?
> > 3. This is more of a comment than a question, but in response to the last comment (i.e., why RNNs may perform better than other models for continuous BMI). I think theoretically, there is no obvious reason to pit RNNs/LSTM type models against models such as transformers. Transformers can be recurrent too -- see for example, the Energy Transformer (Hoover et al. 2023). There are other, related models, such as an autoregressive Perceiver (transformer based) with a reccurent hidden state that may perform well. If indeed, the theoretical reason RNNs have so much greater performance is the maintenance of a hidden state, these class of models may be relevant for future model evaluations for BMI.

---

> > > ### Author Response · Authors · 2023-08-17
> > >
> > > Thanks for the follow-up questions.
> > >
> > > **Why does the GRU have more parameters than the LSTM?**
> > >
> > > During hyperparameter optimization we allowed the RNNs to have 1-3 layers. The best GRU used 2 layers while the best LSTM used only 1 layer, which explains why the GRU had more parameters.
> > >
> > >
> > > **Isn't it possible that the number of parameters in the feedforward class of models makes it difficult for it to train in the closed-loop (online) setting?**
> > >
> > > In this work, all models were trained using the initial offline dataset before being tested online. Since no updates were performed online, we were able to collect a relatively large amount of training data that approximately saturated performance for each model. However, in training schemes where the model is actively updated in the closed-loop setting (for example in reinforcement learning) it is possible that the higher number of parameters could make training difficult. This point may be relevant when training decoders for paralyzed humans, in which training data may be scarce and decoders may be recalibrated (retrained) multiple times using the online data.
> > >
> > > **Other types of models with a hidden state may be relevant for future model evaluations for BMI.**
> > >
> > > Thanks for the comment and bringing our attention to other interesting recurrent models. Here we aimed to evaluate a few basic model architectures (feedforward, recurrent, and linear) and then explored the memorization ability of recurrent models for closed loop BMI. As you mention, future work may investigate these other recurrent architectures, which may enable further accuracy improvements. To better address this limitation, in our global response we proposed adding to the discussion, “More optimal architectures likely exist, but here we found several key factors of each class (for example, the need for limited time history of feedforward models, and the ability of RNNs to strongly learn few movements).”

---

> > > > ### Comment · Reviewer_WBvg · 2023-08-17
> > > >
> > > > I thank the author for the responses. I raise my score from 5 to 6.
> > > >
> > > > While I do think some of the main claims of the paper (e.g., balance of memorization vs generalization) might be limited to the type of BMI task they perform and make it hard to assess how generally applicable those claims are, the authors have done a fine job in mentioning these limitations. One last comment that I do think would be important to include somewhere in the manuscript is that the dichotomies of RNNs v. transformers are not mutually exclusive. There are likely components in each of these type of models that may be complementary in an online BMI setting (e.g., self attention can be combined with recurrence + gating).

---

> > > > > ### Author Response · Authors · 2023-08-20
> > > > >
> > > > > We appreciate the score update. To better address the complimentary nature of RNNs and transformers we propose adding a sentence to the limitations paragraph:
> > > > >
> > > > > “There are several limitations to the present study of closed-loop BMI performance. First, we did not explore every possible decoder architecture. More optimal architectures likely exist, but here we found several key factors of each class (for example, the need for limited time history of feedforward models, and the ability of RNNs to strongly learn few movements). *Future work may explore models that combine recurrence with feedforward attention layers to take advantage of the benefits of each (for example, Jaegle et al. 2021).* Also, only one monkey was used for the online analyses; additional users may have different control strategies or decoder errors.”

---

### Official Review · Reviewer_3c6F · 2023-07-05

**Soundness:** 3 good
**Presentation:** 4 excellent
**Contribution:** 3 good
**Rating:** 7
**Confidence:** 4

**Summary:**

This paper uses an RNN for motor control task. Specifically, it uses the RNN to decode finger movements using neural data recorded from NHPs.

**Strengths:**

The following are the strengths of this paper:
-Online decoding using RNNs.
-Evaluation and comparison with other neural network algorithms.
-Evaluation with higher channel counts.
- Recovering performance using RNNs.

**Weaknesses:**

-There have been several publications including previous NEURIPS papers that show use of RNNs for decoding neural data.
-The algorithms are mostly used as is. The novelty and strengths of the manuscript lies in the system engineering.

**Questions:**

I have following questions.
- SBP vs spike thresholding. There was a single reference that was cited. Do you see a marked difference between the two. If SBP gives similar correlations then it would be highly beneficial from hardware perspective since the sampling rate can be dropped to 2KSPS?
- How is higher channel count simulated? Is there a simulator that enables generating BMI data? Can you simulate and generated a longer dataset (more sessions) for training with same channel count (Does that have benefit on using RNN vs KF ).
- Figure 3 and Section3.2 are the decoders retrained for different targets? For instance will you retrain/calibrate the decoders for different set of target or are you having a constant training set and then just re-evaluating the decoder for multiple targets?
- For all the decoder are they trained and tested on the same session?



**Limitations:**

Check weakness and questions.

---

> ### Author Rebuttal · Authors · 2023-08-09
>
> We thank the reviewer for their questions.
>
>
> **Do you see a difference between SBP and spike thresholding?**
>
> In this work we did not directly compare SBP to spike thresholds but used SBP as the feature for all decoders. Anecdotally, in experiments prior to this study, Monkey N was unable to complete trials using a threshold-based Kalman filter but could successfully perform trials using an SBP-based Kalman filter. This is likely due to visibly low SNRs for most channels on Monkey N’s array, which may be more easily decoded using SBP than thresholds (Nason et al. 2020).
>
> As you mention, SBP can substantially reduce the sampling rate (to 2 KSPS) for a large reduction in power (Irwin et al. 2015, Nason et al. 2021, Costello et al. 2023). Thus, because of the strong decoding performance and the ability to use SBP on battery-powered implantable systems, we used SBP for the decoder comparisons here.
>
>
>
> **How is the higher channel count data simulated? Can you generate more data?**
>
> We appreciate the reviewer pointing out the lack of explanation for simulated data. This information was in the supplement, so we propose adding another methods section as noted in the Global Response.
>
> Briefly, we simulated data using a virtual user who performed the task by moving to targets, and then generated artificial channels where each channel had a random relationship with the virtual user’s position, velocity, and acceleration. This method was based on Trucollo et al. 2008, and is similar to random tuning methods used in other work (Churchland et al. 2012, Willet et al. 2019). Here, the purpose of the simulation was to simply generate datasets with any length, number of channels, number of degrees-of-freedom, or task complexity, which could then be used to measure relative decoder performance across these varied parameters (for example, Fig 2c and Supplemental Fig 3). However, since the simulated channels are randomly generated and do not match the real neural channels, they cannot directly be used for generating additional training data or extrapolating absolute performance to higher channel counts (although there are methods for generating additional realistic datasets).
>
>
>
> **Are decoders retrained for different targets? Are decoders trained/tested in the same session?**
>
> For each different number of targets we collected a training set and trained a decoder on that specific setup. This means each datapoint in figure 3 is a fully separate decoder. The only exception to this is section 3.4 where we purposefully tested the ability of a decoder to generalize to more targets.
>
> All decoders were trained and tested within the same session. While there are methods of incorporating previous datasets into training (like using a day-specific input layer) or recalibrating previous decoders, here, using able-bodied monkeys, we could obtain enough daily training data to saturate performance without added complexity.
>
> To help clarify these points, we propose adding the following:
> Results 3.1, line 221, “Decoders were trained using only same-day data.”
> Results 3.2, line 252, “Separate decoders were trained for each number of targets.”

---

> > ### Comment · Reviewer_3c6F · 2023-08-16
> >
> > Thank the authors have answered my questions. I am updating my decision.

---

### Author Rebuttal · Authors · 2023-08-09

We thank the reviewers for their support of this work and their helpful questions and suggestions. Our work demonstrates state-of-the-art control of a finger brain-machine interface, where we show the importance of closed-loop decoder testing and benefits of decoder memorization. The majority of reviewer comments were clarification questions and suggestions for further discussion, which we have addressed both here and in the individual comments.

**Amount of Time history for Feedforward Models**

Multiple reviewers mentioned that the convolutional and transformer models may have lower performance due to using a time window of only 5 bins (160ms). For the transformer, during initial testing we found that using longer context windows of 10-20 bins (320-640ms) resulted in higher offline correlations, but the decoder was unusable during online control (the decoder worked for 1-2 trials before becoming stuck and was unable to move). Using a shorter window (~5 bins) resulted in better online control where the decoder did not get stuck or biased. We found a similar result with the convolutional network which had maximal closed loop performance with ~160ms of history and performance dropped as more history was added. This suggests that the feedforward models may tend to overfit to the hand-control training data and pay too much attention to earlier bins, showing the benefits of closed-loop testing. These problems could potentially be mitigated by training on more erroneous/imperfect movements or discouraging attention to earlier bins, but further experiments are needed.

We propose adding the following paragraph to the discussion (line ~341):
“In additional testing we found that using more time history in the FNN and TFM decoders (up to 20 bins/640 ms) resulted in poor online control. This is likely due to overfitting to training data that lacks many of the corrective movements present in online control, and demonstrates how offline analyses can fail to predict the performance in the closed loop setting. Interestingly, the RNNs were trained with longer history but did not display this overfitting. Future work may investigate methods of using longer training sequences while limiting overfitting, for example by discouraging attention to early bins.”

**Generalizability to more complex tasks**

Multiple reviewers suggested discussing the ability of a “memorized” decoder for a simple task to generalize to more DoFs or other tasks, for more naturalistic BMI. First, we note that prosthetics and human interfaces can be highly functional with only 2-3 degrees of freedom. For example, a prosthetic hand with control of the index/thumb “pinch” and MRS fingers open/close can form many different hand grasps, or a 2d cursor can control a computer.

Our results suggest that the level of memorization can be controlled within each DoF independently (results 3.3) and that as more DoFs are added, slightly more training data or channels are needed to maintain accuracy (supplemental fig 3). Thus, we hypothesize that higher DoF tasks will need more data but can be trained and tuned using the same approaches used here. In future BMIs with more channels, the decoder may have high accuracy for fully continuous control without the need for memorization, for a larger output space and more naturalistic control.

We propose adding the following text to the discussion, line 353:
“As BMIs extend to more DoFs, our results suggest that more data or neural channels will be needed to maintain accuracy, and the degree of memorization can be controlled for each DoF independently.”


**Simulated Data Methods**

We propose adding the following methods section to better describe the simulated data. This information was in the supplemental methods but not referenced in the main text:

“For some analyses we created simulated offline datasets of a virtual user performing the same target acquisition task. The goal of these simulations was to test the relative impact of amount of training data, number of DoFs, and number of inputs on decoder performance, rather than measuring absolute performance. The simulated user moved with a velocity proportional to the distance to the target along each DoF. Artificial neural channels were generated such that each channel had a random relationship with the position, velocity, and acceleration at each timestep, as suggested in Trucollo et al. 2008. Further details and specifics can be found in Supplementary Methods.”



**Limitations – inability to explore the full parameter space**

Based on multiple reviewers pointing out limitations, we propose adding the following paragraph to the discussion:

“There are several limitations to the present study of closed-loop BMI performance. First, we did not explore every possible decoder architecture. More optimal architectures likely exist, but here we found several key factors of each class (for example, the need for limited time history of feedforward models, and the ability of RNNs to strongly learn few movements). Also, only one monkey was used for the online analyses; additional users may have different control strategies or decoder errors.”

---

### Decision · Program_Chairs · 2023-09-21

**Decision:**

Accept (spotlight)

**Comment:**

A comprehensive comparison was made for BMI decoding and finds great performance of RNNs. Many engineering challenges were tackled to enable a high performance system. Interesting observations were made, as one reviewer summarizes nicely, “…varying the number of discrete targets allows to interpolate between the behavior of a classifier and a continuous movement decoder, where for few targets, most of the actual movement to the target is memorized and with more and more targets, more of the movement is decoded…” This paper is of potential interest to a broad audience of NeurIPS.